# Symptom-Based Predictive Model of COVID-19 Disease in Children

**DOI:** 10.3390/v14010063

**Published:** 2021-12-30

**Authors:** Jesús M. Antoñanzas, Aida Perramon, Cayetana López, Mireia Boneta, Cristina Aguilera, Ramon Capdevila, Anna Gatell, Pepe Serrano, Miriam Poblet, Dolors Canadell, Mònica Vilà, Georgina Catasús, Cinta Valldepérez, Martí Català, Pere Soler-Palacín, Clara Prats, Antoni Soriano-Arandes

**Affiliations:** 1Barcelona School of Informatics, Universitat Politècnica de Catalunya (UPC⋅BarcelonaTech), 08034 Barcelona, Spain; jesus.maria.antonanzas@estudiantat.upc.edu (J.M.A.); cayetana.l.cano@gmail.com (C.L.); mireia.boneta@hotmail.com (M.B.); cristina.aguilera.gonzalez@estudiantat.upc.edu (C.A.); 2Department of Physics, Universitat Politècnica de Catalunya (UPC⋅BarcelonaTech), 08028 Barcelona, Spain; aida.perramon@upc.edu (A.P.); mcatala@igtp.cat (M.C.); clara.prats@upc.edu (C.P.); 3ABS Borges Blanques, Institut Català de Salut (ICS), 25400 Lleida, Spain; capbert@gmail.com; 4Equip Pediatria Territorial Alt Penedès-Garraf, Institut Català de Salut (ICS), 28036 Barcelona, Spain; annagatellcarbo@gmail.com (A.G.); pepepediatre@gmail.com (P.S.); cvalldeperez@gmail.com (C.V.); 5Equip Territorial Pediàtric Sabadell Nord, Institut Català de Salut (ICS), 08206 Barcelona, Spain; miriampoblet@gmail.com; 6CAP Barberà del Vallés, 08210 Barcelona, Spain; dcanadell03@gmail.com; 7EAP Horta, 08024 Barcelona, Spain; mvilad.bcn.ics@gencat.cat; 8CAP Drassanes, 08001 Barcelona, Spain; gcatasus.bcn.ics@gencat.cat; 9Comparative Medicine and Bioimage Centre of Catalonia (CMCiB), Fundació Institut d’Investigació en Ciències de la Salut Germans Trias i Pujol (IGTP), 58525 Badalona, Spain; 10Paediatric Infectious Diseases and Immunodeficiencies Unit, Hospital Universitari Vall d’Hebron, 08035 Barcelona, Spain; psoler@vhebron.net

**Keywords:** machine learning, deep learning, paediatrics, SARS-CoV-2, COVID-19, epidemiology, microbiology

## Abstract

Background: Testing for severe acute respiratory syndrome coronavirus 2 (SARS-CoV-2) infection is neither always accessible nor easy to perform in children. We aimed to propose a machine learning model to assess the need for a SARS-CoV-2 test in children (<16 years old), depending on their clinical symptoms. Methods: Epidemiological and clinical data were obtained from the REDCap^®^ registry. Overall, 4434 SARS-CoV-2 tests were performed in symptomatic children between 1 November 2020 and 31 March 2021, 784 were positive (17.68%). We pre-processed the data to be suitable for a machine learning (ML) algorithm, balancing the positive-negative rate and preparing subsets of data by age. We trained several models and chose those with the best performance for each subset. Results: The use of ML demonstrated an AUROC of 0.65 to predict a COVID-19 diagnosis in children. The absence of high-grade fever was the major predictor of COVID-19 in younger children, whereas loss of taste or smell was the most determinant symptom in older children. Conclusions: Although the accuracy of the models was lower than expected, they can be used to provide a diagnosis when epidemiological data on the risk of exposure to COVID-19 is unknown.

## 1. Introduction

The severe acute respiratory syndrome coronavirus 2 (SARS-CoV-2) pandemic continues to be a priority health problem worldwide. More than eighteen months after the beginning of the COVID-19 pandemic, as of 15 October 2021, more than 239 million cases have been reported, with more than 4,886,000 associated deaths worldwide [1]. 

Children and adolescents are also susceptible to being infected by SARS-CoV2. According to data from the Catalan Agency for Health Quality and Evaluation (AQuAS) [2], since the beginning of the pandemic, there has been 191,412 (end of September 2021) confirmed cases in children and adolescents (0–17 years), accounting for 19.2% of the total cases reported to the Catalan Health Surveillance System [2]. Most children have a mild symptomatic disease. According to the available data, children with COVID-19 have a better prognosis than adults, and mild cases recover within 1–2 weeks from the onset of the disease [3]. A prospective observational study carried out in Catalan households in Summer 2020 [4] reported that half of the paediatric cohort (47.2%) was asymptomatic [4], compared to around 40% in the general population [5]. The same study reported that 71.9% of symptomatic children presented fever, 37.4% cough, 23.6% headache, and 23.2% fatigue [4]. However, these clinical manifestations are nonspecific to SARS-CoV-2, mainly in children who can be affected by other respiratory or gastrointestinal viral infections. Therefore, there is a need to find decision-making support tools that assist in providing a more accurate diagnosis. Children, paediatricians and public health policy-makers could benefit from knowing which presenting symptoms are most likely to be associated with SARS-CoV-2 infection. This could result in a reduction in the number of diagnostic tests for SARS-CoV-2 infection in future.

The use of algorithms based on clinical data has been reported as a useful tool for clinical decision-making in emergency departments, to avoid work overload and to improve the quality of care [6,7]. In fact, predictive models that combine patient and disease characteristics to estimate the risk of a poor outcome from COVID-19 have already been published [8]. However, a systematic review showed the little value that signs and symptoms have on the accuracy of the SARS-CoV-2 diagnosis in adults, and the authors emphasised the urgent need for prospective clinical studies to evaluate this issue [9]. In children and adolescents, a systematic review of symptoms and signs of COVID-19 found that fever and cough were the most common symptoms; however, the authors stated that further research with community-based cases is needed to properly identify and test children and young people for this disease [10].

The characterisation of diseases, such as COVID-19, in children using classical data analysis techniques, is complex, because in most cases the data are high-dimensional (each patient generates a considerable amount of information or variables). In addition, there may be interactions between variables that make prediction a complicated task. Taking all of these into account, we aim to evaluate and establish the diagnostic performance of the symptoms and signs (isolated and in combination) in community-based children with suspected COVID-19, as a screening tool to rule out the disease (diagnostic model). We used machine learning and deep learning techniques (artificial intelligence) to model the complex interactions between different symptoms, and we applied new interpretability techniques that can be used to understand both the symptoms that affect the whole population and those that determine the probability of individual infection.

## 2. Materials and Methods

### 2.1. Study Design

This is a prospective, observational, cross-sectional study of symptomatic children <16 years old with a suspected COVID-19 disease, who were tested with rapid antigenic diagnostic tests (RDT), reverse transcription-polymerase chain reaction (RT-PCR) or both for confirmatory diagnosis in any of the participating Primary or Hospital health-care centres between 1 November 2020 and 31 March 2021, following the standard of care for diagnosing these cases.

### 2.2. Sampling

RDT (PANBIO COVID-19 Ag rapid test device, Abbott©) and RT-PCR for SARS-CoV-2 were administered in Primary Health Care Centres of Catalonia (Spain), as per the protocol published by the Catalan Society of Paediatrics (www.scpediatria.cat, accessed on 17 November 2021) [11]. In the participating hospitals, any suspected case was tested for SARS-CoV-2 or other respiratory viruses to confirm the diagnosis.

We followed the STROBE statement for observational studies [12].

### 2.3. Data Sources and Setting

Catalonia, an autonomous region in northeast Spain with 7.5 million inhabitants (1,225,707 younger than 16 years of age), has a universal, publicly-funded health system with 7 sub-regional departments and more than 400 primary health care centres. Within the COVID-19 Paediatric Research Group in Catalonia (COPEDI-CAT) project, more than 140 paediatricians from primary health centres and public and private hospitals prospectively collected demographic, epidemiologic, clinical, and diagnostic data on suspected respiratory viral infection (RVI) cases. Along with this data collection, and to determine the epidemiological trend of COVID-19 in this setting, information on the total and positive SARS-CoV-2 RT-PCR results involving eligible participants was delivered by the Catalan Agency for Health Quality and Evaluation (AQuAS), which obtained the data from the Catalan Epidemiological Surveillance Network and the referral microbiological laboratories. 

During this study period, the schools remained open without restrictions, but applied strict non-pharmaceutical interventions [13], including face masks in classrooms and school buildings for children over the age of 6.

This study was conducted in a context of high community circulation of the SARS-CoV-2 virus, with a total of 337,271 cases reported by the Catalan surveillance system among the general population, and 45,914 among children below 16 (13.61%). During the first period of the study (November–February), the predominant variant was B.1.177. The transition to the Alpha variant started in December 2020, when the first cases were detected by the sequencing surveillance system [14], and represented more than 50% of the cases from mid-February to the end of the study. 

Overall, we collected demographic, epidemiological and clinical and microbiological data at the time of diagnosis or test performing, on 4456 symptomatic children with suspected SARS-CoV-2 infection (see Appendix A).

### 2.4. Case Definition

A confirmed COVID-19 case was defined as any individual testing SARS-CoV-2 positive by real-time RT-PCR or by RDT in a respiratory specimen. 

### 2.5. Recruitment Process

To avoid selection bias in case recruitment, paediatricians recorded all the suspected cases seen in their daily practice. However, during work overload peaks, they only collected data from the first 5 suspected cases per day. Follow-up was performed by the patient’s paediatrician during a primary care visit or by a telephone interview with the parents or legal guardians, using the planned questionnaire. All data were recorded in a web-based platform, the Research Electronic Data Capture (REDCap^®^) database. 

### 2.6. Ethical Considerations

Ethical approval was obtained from the referral IDIAP J. Gol Research Foundation for Primary Care in Catalonia, Spain (20/187-PCV), and the coordinating centre of the study, Vall d’Hebron Research Institute, Barcelona, Spain (PR(AG)475/2020) on 25 September 2020. 

### 2.7. Data Description

The epidemiological and clinical data of the registered cases are described in Table 1. The main objective of this study was to determine which symptoms are decisive within the paediatric-age population to define whether a PCR test should be performed when a child presents symptoms suggestive of COVID-19. Therefore, only symptom-related variables were analysed in this study. Before pre-processing the data to train the predictive models, a statistical analysis was performed to determine the weight of each of the clinical features in the dataset for the final diagnosis, to avoid using uninformative variables. As the distribution of the data was unknown and the variables were independent, we performed a chi-square (χ^2^) test, considering statistically significant any *p*-value < 0.05. For the analysis to be relevant, we only consider patients that underwent a COVID-19 test and have a reported positive or negative result from it, leading to a cohort of 4419 individuals for the statistical analysis.

The co-viral and bacterial infections were diagnosed upon specific testing under clinical suspicion. Bacterial infection was defined when culture growing (haemoculture or culture of the specimen) or bacterial PCR tested positive. The comorbidities that have been included are congenital cardiopathy, hypertension, asthma, chronic pulmonary disease, renal, liver or neurologic disease; diabetes mellitus, tuberculosis, primary and secondary immunodeficiencies (including human immunodeficiency virus (HIV) infection), onco-haematological disease, Kawasaki syndrome, auto-inflammatory diseases, obesity, prematurity and palivizumab administration due to prematurity. In addition, the social data have been obtained due to the effort of the researchers in fulfilling the planned questionnaire in the different paediatric practices.

### 2.8. Pre-Processing

To obtain homogeneous labelling of the data, a thorough pre-processing was conducted. Initially, the diagnosis was coded with three possible values: COVID-19 positive, COVID-19 negative or suspected virus infection. However, for this last option, we checked whether PCR or RDT for SARS-CoV-2 was performed, and based on the result of these tests, the child was classified as positive or negative for COVID-19. If a child received both tests, priority was given to the result of the PCR over the RDT. 

Most of the clinical variables were coded as binary using “1” or “0” values, depending on whether the symptom was present or not, respectively. In addition, fever was coded as slight, moderate or high, with “0” meaning no fever, “1” for 37.5 to <38 °C, “2” for 38 to 39 °C, and “3” for >39 °C. The total days of fever were defined as “0” for no fever, “1” if the patient had fever 1 or 2 days, “2” for fever lasting 3 to 7 days and “3” for the presence of fever more than 7 days. Lastly, auscultation was encoded such that “0” means no pathologic auscultation, “1” stands for wheezing, “2” for crackles and “3” for both.

Some data were missing in certain variables, mostly due to the age of the patients. In fact, some of the symptoms are a child’s self-reported characteristics, such as headache or loss of smell and taste. In younger children aged from 0 to 5 years, we were unable to obtain certain clinical information due to the lack of communicative skills, intrinsic to that age.

Symptoms with a percentage of missing data in excess of 25% were eliminated from the models, as explained in the Appendix A as they did not provide clear information about SARS-CoV-2 infection in children. 

Besides, as a different pattern of symptoms was expected to be found according to the age of the child, the general training set was divided into two subsets to train age-specific predictive models for SARS-CoV-2 infection. Thus, three models were developed, one for the general paediatric population (0 to <16 years), and two for children aged 0 to 5 years and 6 to <16 years, respectively. Using age-related data subsets, we were able to rule out certain variables with a high number of missing values, but still keep enough data for the models to be trained in a balanced way (50% positives and 50% negatives) to induce generalisation capabilities. Hence, 1540 patients were considered for the general model, 448 for the model for children ages 0–5, and 1026 for the model for children ages 6–15.

Table 2, Appendix A show the final symptoms that were chosen to train the general paediatric model, for children under 5 and children aged 6–15 years, respectively. Besides, it is shown how many patients presented each symptom, and how many of those tested positive for COVID-19. For example, the subset of data to train the 0 to 5-year-old children model, in the Appendix A, does not use the absence of taste/smell symptoms, since this information is not straightforward to obtain from the younger children. 

### 2.9. Methodology Implementation

We developed a systematic pipeline to obtain each predictive model as we implemented several machine learning (ML) and deep learning (DL) architectures to determine which outperforms the others for each data subset (see Appendix A). 

We followed a pipeline (Figure 1) that is broken down into data processing, model selection, fine-tuning and evaluation steps. Given a particular model architecture, relevant data and hyperparameter configurations to test (Appendix A), the pipeline outputs the best hyperparameter configuration, as well as some evaluation metrics. Further description of the pipeline can be found in the Appendix A.

### 2.10. Model Development

We divided each data subset into training and testing sets in 70/30 proportions. The models were cross validated (CV) to evaluate their performance. We trained the candidate model architectures with several hyperparameter configurations and tested their performance against the cross-validation sets. Fine-tuning was then used to optimise the chosen set of hyperparameters. Average validation scores for each of the chosen configurations and an overview of the tuned hyperparameters and the number of configurations tested are reported in Appendix A. The performance was quantified in terms of area under the receiver operating characteristic curve (AUROC), sensitivity, specificity, precision and F1 score. These terms are defined in the Appendix A.

### 2.11. Feature Importance Extraction

We computed SHAP (SHapley Additive exPlanations) [15] values for all test instances to understand the global importance of each of the variables on the final classification done by the implemented model. SHAP is a model-agnostic explainer method in which, for a particular instance of a dataset, and given the model’s classification prediction for that instance, each feature is assigned a weight in terms of how much it impacted the model’s output with respect to the expected output. 

While we were only able to interpret the model trained with data from all age groups and then analyse the SHAP values of observations by different age strata instead of creating a classifier for each age interval, we believe the latter yields a better interpretation. That is, in the first case, the SHAP values of observations in one age stratum may possibly be influenced by observations of other age groups, with the results being biased depending on what the classifier actually learned. On the other hand, having separate models ensures that the feature importance of each age stratum is not influenced by leading factors in other age groups. We are thus able to see how the predictors influence the output over different age ranges. We used the implementation from [15].

## 3. Results

### 3.1. Data Description

A total of 4456 children were recruited in our study, 44.5% (1984/4456) were female and 42% (1872/4456) were younger than 6 years of age, with no significant differences between COVID-19 and non-COVID-19 cases (Table 1). Diagnostic tests for SARS-CoV-2 (PCR and/or RDT) were administered in 4434 (99.5%) of the total recruited cases (Table 1). Of the 840 cases with a PCR test and 3916 with RDT, 321 (38.2%) and 463 (11.8%) tested positive, respectively. Both diagnostic tests (PCR and RDT) were performed at the same time in 354 children: 14/354 tested positive for both (4%); 108 (30.5%) yielded discordant results, 89 (25.1%) with positive PCR and negative RDT, and 19 (5.4%) with negative PCR and positive RDT; and the remaining cases were negative for both tests (232/354, 65.5%). Among children testing negative for SARS-CoV-2 either with RDT or PCR, we found one case of influenza A and 19 cases of adenovirus with RDT and 10 cases after performing a multiplex PCR: rhinovirus (*n* = 3), adenovirus (*n* = 3), another coronavirus (*n* = 1), enterovirus (*n* = 1), virus Epstein-Barr (*n* = 1), and bocavirus (*n* = 1). No cases of respiratory syncytial virus or influenza B were detected. Co-viral infections with SARS-CoV-2 were two rhinoviruses and one case of enterovirus. The use of the school bus, athletic activities, and suspected or confirmed COVID-19 cases at home or school was associated with COVID-19 diagnosis (Table 1). No significant differences were observed between COVID-19 and non-COVID-19 cases for comorbidities. 

The results of the χ^2^-test are shown in Appendix A. The degree of fever was relevant to determine whether the patient had COVID-19 or not (Appendix A). A higher fever was associated with a SARS-CoV-2 negative result, while a lower fever may be related to COVID-19. Besides, this descriptive analysis showed that the lack of sense of taste and smell was associated with a non-COVID-19 diagnosis, in contrast to what is shown in adults. Despite that, we should note that the number of patients reporting a lack of smell and taste is low compared to the whole sample, which is a strong bias to take into account when understanding this analysis. It is also important to note that some symptoms, such as confusion, correlated strongly with the COVID-19 diagnosis due to the high number of missing values or/and to a low number of affected patients, meaning they had to be discarded from any present and future analysis.

### 3.2. Model Development

In this section, we report the scores of the best performing classifiers for each data subset. Table 3 contains the average CV scores of the best configuration found for each architecture tested, while Table 4 contains the test scores for the final fine-tuned architectures with 95% confidence intervals. Table 3 shows the tied results between random forest (RF) and kernel support vector machine (kSVM). We chose RF as the best architecture because we were able to compute its exact SHAP values efficiently, instead of approximating them.

We assessed discrimination by quantifying the AUROC. All the architectures performed quite similarly to each other in each subset, with Boosted Trees (XGB) being the best for the subset including all ages (AUROC = 0.65), and RF the best for subsets aged 0 to 5 and 6 to <16 years, with an AUROC of 0.63 and 0.67, respectively (Table 4). Classifiers performed the worst in the subset for ages 0 to 5 years, while the scores in the subset for all ages and 6 to <16 years were very similar. This could mean that either there were not enough observations in the 0–5-year subset for the classifiers to properly learn patterns, or there were no relevant symptom patterns to be learned. This latter hypothesis is supported by the fact that performance in the subset for all ages was worse overall than in the subset for ages 6 to <16 years, the latter being a subset of the former, which means that adding the 0–5-year subset to the 6 to <16 years subset only confuses the classifiers. This is in agreement with the common low specificity of clinical characteristics observed in younger children.

### 3.3. Feature Importance Extraction

The figures with multiple colours in this section are beeswarm plots. They are ordered by decreasing overall importance from the top to the bottom of the figure. For each feature, the SHAP value of each test observation is shown as a point. The observation has the symptom defined by the feature if the colour is red, and does not have it if it is blue. The more to the left the points are, the more the output is associated with an absence of SARS-CoV-2 infection. The more to the right the points are, the more the output is associated with a SARS-CoV-2 infection.

#### 3.3.1. General Model

In the model trained for the dataset containing patients aged 0 to 15 (the whole set) (Figure 2), the presence of headache and fatigue positively influenced the likelihood of infection, while the presence of odynophagia, vomiting or diarrhoea negatively influenced it. Moreover, developing fever for 1–2 days, or fever higher than 39 °C, or wheezing and nasal congestion were also associated with a lower probability of SARS-CoV-2 infection. On the contrary, mild fever (38 to 39 °C) or loss of smell or taste increased the likelihood of COVID-19. 

Figure 3A,B show the relative importance of each variable. For example, on average, reporting a headache has approximately twice the effect on the model’s decision compared to vomiting. This figure does not take into account if features affect the predictions positively or negatively, but it can help the decision-making process by providing a hierarchy of importance. Figure 3B shows the maximum impact that a clinical characteristic had on the prediction of a test observation. This differs from the average impact, and it underscores how a diagnosis process based on a contextual population needs to be individualised for each patient. For example, although the average impact of the loss of smell was +0.09, its maximum impact was +1.47, five times higher than the most impactful feature. Without considering the magnitude of the maximum impact, almost all the top-10 average features (Figure 2 and Figure 3A) were also the top-10 maximum impact features (Figure 3B), except nasal congestion and fever lasting for 1–2 days.

#### 3.3.2. Model for Children by Age Range

In the model trained for the dataset containing patients aged 0 to 5, having fever from 3 to 7 days, fever higher than 39 °C, odynophagia, visible skin rashes, shortness of breath, wheezing and fatigue all impacted negatively on the predicted likelihood of SARS-CoV-2 infection (Figure 4A). On the other hand, low fever (37.5 to 38 °C), cough and fever for 1 to 2 days were associated with a higher probability of SARS-CoV-2 infection.

In the model trained with data from patients aged 6 to 15 (Figure 4B), the presence of painful swelling, vomiting, diarrhoea, wheezing and gastrointestinal symptoms were associated with a lower infection probability. On the other hand, loss of taste and smell, cough, headache and mild fever (from 38 to 39 °C) contributed, on average, to the likelihood of infection.

In general, we noticed how the absence of a symptom did not negatively influence the predicted probability of whether a child is infected with SARS-CoV-2. Instead, the SHAP explanation method captured that the model weighted the presence of a symptom for a positive or negative prediction.

Odynophagia was present as a relevant factor for non-COVID-19 diagnosis in all models. Low and mild fever (37.5–39 °C) was the most relevant sign for a positive diagnosis for children under 6 years of age. A high fever (>39 °C) was likely related to a non-COVID-19 diagnosis in the majority of cases, as well as vomiting and diarrhoea (at least for children aged >6). Headache and fatigue were relevant symptoms for a COVID-19 diagnosis in the overall model (although the former cannot be applied to children aged <6). Overall, mild fever (38–39 °C) was indicative of a COVID-19 diagnosis, and high fever (>39 °C) of a non-COVID-19 case. Vomiting and diarrhoea were associated with a non-COVID-19 diagnosis in the models for all ages and children aged >5 years but was not relevant for younger children. 

We noted the value of the feature importance analysis. In the descriptive data analysis, only a small fraction of children with COVID-19 had a loss of smell and taste. In fact, the χ^2^-test suggested that these alterations were no signs of a COVID-19 diagnosis. However, for the black-box classifiers developed, these alterations were associated with a COVID-19 diagnosis. Rather than a contradiction, this suggests that the ML models capture complex interactions in the data that are not straightforward in a simple descriptive analysis.

## 4. Discussion

### 4.1. Main Results

The use of ML and DL techniques (artificial intelligence) to model the complex interactions between different symptoms in children with a suspected SARS-CoV-2 infection demonstrates the challenges of combining clinical characteristics to predict a COVID-19 diagnosis in children (AUROC = 0.65), especially in those younger than 6 years of age (AUROC = 0.63). Low-grade fever was the major sign to predict COVID-19 in younger children, whereas loss of taste or smell was the most determinant symptom in older children. 

The models’ accuracy was not very high, which is in agreement with paediatric low specific characteristics of COVID-19 symptomatology in children. Nevertheless, the models were capable of identifying some patterns that are easy to see from a descriptive approach. It is worth noting that the results for older children were more accurate than for younger ones, which was to be expected since older children can better describe their symptoms, which makes the data more reliable. Another factor is the non-specificity in children aged <6 years of other common respiratory viruses that children suffer from in the first years of life.

Therefore, these models could serve as a clinical tool to support the decision to administer a diagnostic test such as PCR for SARS-CoV-2 to confirm a diagnosis in a given epidemiological context, in combination with the paediatrician’s criteria. 

Before processing the data with predictive models, a conventional statistical analysis showed that the use of the school bus, playing sports, and suspected and confirmed COVID-19 cases at home or school were all associated with a COVID-19 diagnosis (Table 1). Thus, there are epidemiological characteristics linked to the risk of exposure for SARS-CoV-2 that are key when deciding whether to perform a diagnostic test to rule out COVID-19. However, the model can be of use when there is no accurate information about close contact with COVID-19, or this is unknown due to potential multiple risk exposures. Moreover, in a future scenario with lower incidences and more relaxed control protocols for SARS-CoV-2, the model will be especially relevant in symptomatic cases that are not included in contact-tracing studies. Therefore, we decided to use ML techniques to explore whether or not a group of symptoms could be a predictor of COVID-19 in children in different age groups.

### 4.2. Comparison with Prior Work

In Israel, a national symptom survey study was carried out among the adult population to build up a prediction model to prioritise individuals for a SARS-CoV-2 test [16]. The result of this study was the development of a tool that could be used worldwide, but mainly in areas with limited testing resources, thereby increasing the rate at which positive individuals could be identified. Moreover, individuals at high risk for a positive test result could be isolated prior to testing [16]. Eighteen self-reported symptoms have been used in a population older than 16 through a mobile application for the early detection of SARS-CoV-2 infection to contain the spread of COVID-19 and efficiently allocate medical resources [17]. Prognostic models to predict the risk of clinical deterioration in acute COVID-19 adult hospitalised cases have been shown to provide a great clinical advantage [18] because they can be easily collected as part of daily routine care. In fact, reliable predictive models can be a means to improve clinical management and, consequently, to better allocate human and economic resources [19]. Other predictive models have been applied with excellent results among laboratory parameters values of patients who died of COVID-19 to determine the risk of mortality [20]. Additionally, systematic evaluation of different prognostic models among hospitalised adults with COVID-19 has been successful showing strong predictors of deterioration and mortality in this population [21], or for long-covid symptoms [22]. However, all of these studies were performed among the adult population. In children and adolescents, fever and cough were the most common symptoms in a systematic review of more than 1300 studies, with the same clinical characteristics observed in most of the other respiratory viral infections [10]. 

Research involving prognostic factors in paediatric populations with SARS-CoV-2 infection has primarily focused on hospitalised children [23,24], although some studies with a community-based approach have been recently published to assess symptom patterns associated with a positive result for a SARS-CoV-2 swab [25]. These authors found that the symptoms strongly associated with a positive result for a SARS-CoV-2 swab were loss or change of smell or taste, nausea/vomiting, headache and fever [25], whereas other studies identified additional symptoms such as persistent cough, chills, appetite loss, and muscle aches [26], which were also predictive of the Alpha variant SARS-CoV-2 infection. Our main strength is that the cases were recruited from the community by primary healthcare paediatricians, minimising any potential selection bias. As a consequence of this approach, our model is comparable and similar to the above-mentioned community-based studies for the older paediatric age group, but not for the younger children (<6 years old) because of the lack of information about loss of smell or taste is intrinsically associated with their age. In fact, only two of the whole set of symptoms, low fever (37.5 to 38 °C) and cough, were associated with a higher probability of SARS-CoV-2 infection in younger children in our study. Finally, this model could be useful for administrators of schools or day-care centres to consider reassessing the symptoms to include only those that are most strongly associated with positive results for swabs for SARS-CoV-2 infection.

### 4.3. Limitations

Our study has some limitations. Firstly, some missing values limited the training performance of the model learning approach, and greater sample size could have improved the AUROC value. In fact, for data as complex as these, a large number of observations is probably needed for a non-overfitted behaviour and a higher predictive performance. Secondly, the majority of tests administered in the study are RDT, which are, in general, less sensitive than RT-PCR, and this could introduce a bias in the former analysis. However, the RDT used in this study meet the WHO criteria of ≥80% sensitivity and ≥97% specificity, and from our cohort of patients only symptomatic were tested and in their early symptomatic period which increases the diagnostic value, as found in [27]. This would mean that, if there are 20% of patients with a false negative diagnosis, the positive RDT patients would actually increase by 115. That loss in our study is acceptable, as it would not make a significant improvement in the model training. On the other hand, our model was trained for symptoms collected in paediatric cases between November 2020 and March 2021, when B.1.177 (November-February) and Alpha (B.1.1.7) (February–March) were the predominant SARS-CoV-2 variants in Catalonia (Spain), and this analysis could have given different results for new SARS-CoV-2 variants, such as Delta (B.1.617.2). In the UK, a recent study showed that the odds of several symptoms were higher with Delta than Alpha infection, including headache and fever. However, symptoms had a short duration and were similar for both variants, and, fortunately, very few children were admitted to hospital with either variant [28]. Thirdly, we have presented as ‘important features’ those that are only ‘important’ in the context of a unique classifier. Thus, if a classifier has learnt wrong representations from the data, or was simply not able to identify complex patterns, the features presented as important by SHAP will not be representative of reality. 

Moreover, even if the model performance fits the data, feature importance is to be interpreted as correlated to reality and not causal. With this in mind, the better performing the models are, the better feature importance values can be interpreted. In our case, the model for ages 0 to 5 years had the worst performance, so feature importance values are not as representative of the model as for ages ≥6. In addition, SHAP is a ‘permutation-based’ explanation method. Therefore, in order to compute feature importance, some amount of randomisation is required, which entails some degree of instability in the results. If one were to recompute SHAP values for a particular observation multiple times, the results may vary slightly. Despite this, we have used SHAP to represent patterns of the whole data population, not individual instances. Moreover, interactions between factors have not been explored locally (i.e., we determined the weight of each symptom on a positive or negative SARS-CoV-2 test result, but not how different combinations of symptoms may affect the result). 

Besides, although we have characterised each population’s model on average, individual observations can have different features as main contributors to their probability, as can be seen in Figure 3B, which shows the maximum impact of features on any observation. The features with the maximum impact on a model’s output are not ranked the same as the features with the maximum average impact. 

Finally, this study was carried out in a context in which SARS-CoV-2 was circulating freely throughout the community, which increases the chances of positive results, and in the absence of other seasonal viruses such as respiratory syncytial virus or influenza, which may share some symptoms with COVID-19. Therefore, once the seasonal viruses circulate again, comparative studies could be carried out to explore the utility of this methodology in providing a reliable symptom-based diagnosis. 

## 5. Conclusions

We were able to characterise the clinical presentation of the COVID-19 disease in children (<16 years old) with an AUROC of 65%, and also to determine the differences between children <6 years old (AUROC of 63%) and children aged 6 to 16 (AUROC of 67%). 

The present study offers a useful tool for paediatricians to help decide whether to administer a SARS-CoV-2 test or not. In addition, the model could be put in service for the general public by means of, for example, a web or mobile application, and guide the parents or users to decide whether the child should go for a consultation, and hence prevent the collapse of medical institutions.

## Figures and Tables

**Figure 1 viruses-14-00063-f001:**
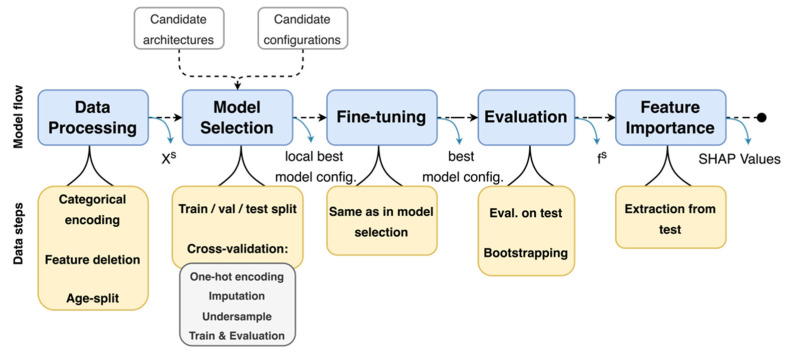
Modelling and data pipeline: a classifier f^s^ and its quality metrics is obtained for each dataset X^s^.

**Figure 2 viruses-14-00063-f002:**
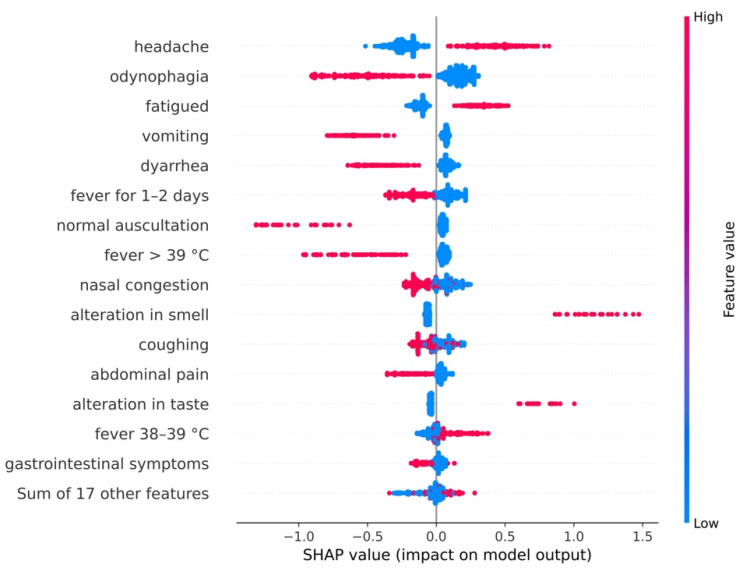
Impact of each variable on the model output for the general model. The features are organised top-down, by decreasing overall importance. For each feature, the SHAP value of each test observation is shown as a point. Each symptom is present if the colour of the point is red and absent if it is blue. The more to the right the points are, the more the output is associated with a SARS-CoV-2 infection.

**Figure 3 viruses-14-00063-f003:**
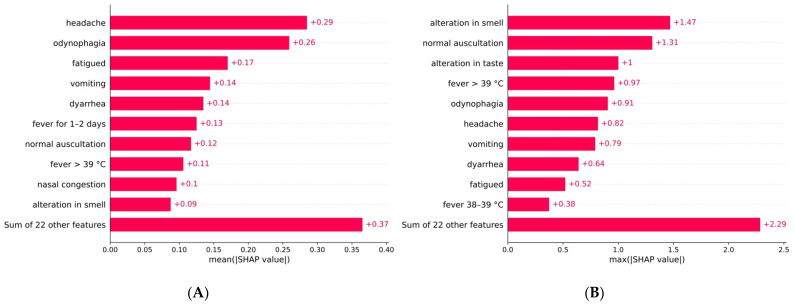
(**A**) Absolute mean impact and (**B**) absolute maximum impact of each variable for the general model.

**Figure 4 viruses-14-00063-f004:**
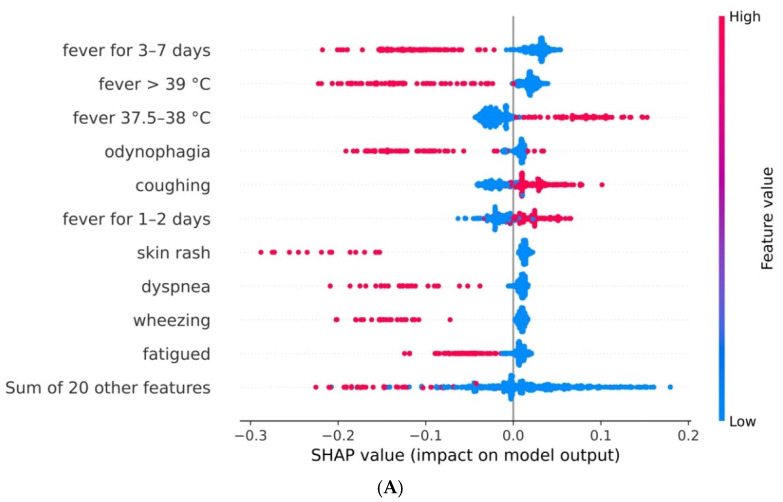
(**A**) Impact of each variable on the model output for the model for children 0 to 5 years old and (**B**) for children 6 to 15 years old. The features are organised top-down, by decreasing overall importance. For each feature, the SHAP value of each test observation is shown as a point. Each symptom is present if the colour of the point is red and absent if it is blue. The more to the right the points are, the more the output is associated with a SARS-CoV-2 infection.

**Table 1 viruses-14-00063-t001:** Baseline epidemiological and diagnostic characteristics of the cases included in the dataset.

Characteristic	N (%)	COVID-19 N (%)	No COVID-19 N (%)	*p*-Value
**Sex (*n* = 4445)**				
Male	2461 (55.4)	428 (55.9)	2033 (55.3)	0.78
Female	1984 (44.6)	338 (44.1)	1646 (44.7)	
**Age (*n* = 4412)**				
0–5	1872 (42.4)	315 (42.1)	1557 (42.5)	0.87
6–17	2540 (57.6)	433 (57.9)	2107 (57.5)	
**Test performed (*n* = 4456)**				
Yes	4434 (99.5)	764 (99.6)	3670 (99.5)	0.99
No	22 (0.5)	3 (0.4)	19 (0.5)	
**PCR result (*n* = 840)**				
Positive	321 (38.2)	321 (100.0)	0 (0.0)	<0.001
Negative	519 (61.8)	0 (0.0)	494 (100.0)	
**RDT result (*n* = 3916)**				
Positive	463 (11.8)	463 (100.0)	0 (0.0)	<0.001
Negative	3453 (88.2)	0 (0.0)	3453 (100.0)	
**X-ray performed (*n* = 4328)**				
Yes	77 (1.8)	15 (2.0)	62 (1.7)	0.54
No	4251 (98.2)	731 (98.0)	3520 (98.3)	
**CT performed (*n* = 4263)**				
Yes	1 (0.02)	1 (0.1)	0 (0.0)	0.17
No	4262 (99.98)	730 (99.9)	3532 (100.0)	
**Use school bus (*n* = 3395)**				
Yes	121 (3.6)	32 (5.5)	89 (3.2)	0.009
No	3274 (96.4)	548 (94.5)	2726 (96.8)	
**Play sports (*n* = 3227)**				
Yes	376 (11.7)	96 (17.1)	280 (10.5)	<0.001
No	2851 (88.3)	465 (82.9)	2386 (89.5)	
**Smokers at home (*n* = 4161)**				
Yes	1232 (29.6)	197 (29.8)	1035 (29.6)	0.93
No	2929 (70.4)	465 (70.2)	2464 (70.4)	
**Persons at home (*n* = 4230)**				
≤4	1360 (32.2)	215 (29.9)	1145 (32.6)	0.17
>4	2870 (67.8)	503 (70.1)	2367 (67.4)	
**Suspected positive at home (*n* = 4456)**				
Yes	956 (21.4)	494 (64.4)	462 (12.5)	<0.001
No	3500 (78.6)	273 (35.6)	3227 (87.5)	
**Confirmed positive at home (*n* = 4456)**				
Yes	548 (12.3)	451 (58.8)	97 (2.6)	<0.001
No	3908 (87.7)	316 (41.2)	3592 (97.4)	
**Suspected positive at school (*n* = 4456)**				
Yes	338 (7.6)	124 (16.2)	214 (5.8)	<0.001
No	4118 (92.4)	643 (83.8)	3475 (94.2)	
**Confirmed positive at school (*n* = 4456)**				
Yes	291 (6.5)	125 (16.3)	166 (4.5)	<0.001
No	4165 (93.5)	642 (83.7)	3523 (95.5)	
**Co-viral infection (*n* = 2306)**				
Yes	14 (0.6)	4 (2.3)	10 (0.5)	0.02
No	2292 (99.4)	171 (97.7)	2121 (99.5)	
**Bacterial infection (*n* = 2363)**				
Yes	60 (2.5)	11 (2.7)	49 (2.5)	0.73
No	2303 (97.5)	390 (97.3)	1913 (97.5)	
**Comorbidities (*n* = 4456)**				
Yes	688 (15.4)	129 (16.8)	559 (15.2)	0.25
No	3768 (84.6)	638 (83.2)	3130 (84.8)	

**Table 2 viruses-14-00063-t002:** Specifications of the dataset used to train the predictive model of COVID-19 in paediatric-aged patients.

Characteristic	Total N (%)	COVID-19 N (%)	No COVID-19 N (%)
**Fever** No 37.5 °C to <38 °C 38 °C to 39 °C >39 °C Unknown	456 (43.51) 208 (19.85) 293 (29.96) 60 (5.73) 31 (2.96)	219 (41.79) 122 (23.28) 141 (26.91) 22 (4.20) 20 (3.82)	237 (45.23) 86 (16.41) 152 (29.01) 38 (7.25) 11 (2.10)
**Cough** No Yes Unknown	574 (54.77) 441 (42.08) 33 (3.15)	303 (57.82) 201 (38.36) 20 (3.82)	271 (51.72) 240 (45.80) 13 (2.48)
**Total days of fever** None 1 or 2 days 3 to 7 days >7 days Unknown	553 (52.77) 368 (35.11) 112 (10.69) 15 (1.43) - (-)	279 (53.24) 188 (35.88) 48 (9.16) 9 (1.72) - (-)	274 (52.29) 180 (34.35) 64 (12.21) 6 (1.15) - (-)
**Auscultation** Normal Pathological Unknown	705 (67.27) 55 (5.25) 288 (27.48)	351 (67.24) 2 (0.38) 169 (32.38)	354 (67.56) 51 (9.73) 119 (22.71)
**Auscultation type** Normal Wheezing Crackles Both Unknown	993 (94.75) 40 (3.82) 6 (0.57) 9 (0.86) - (-)	520 (99.24) 4 (0.76) 0 (0) 0 (0) - (-)	473 (90.44) 36 (6.88) 6 (1.15) 8 (1.53) - (-)
**Dysphonia** No Yes Unknown	971 (92.65) 46 (4.39) 31 (2.96)	486 (92.75) 18 (3.44) 20 (3.82)	485 (92.56) 28 (5.34) 11 (2.10)
**Respiratory sympt.** No Yes Unknown	954 (91.03) 56 (5.34) 38 (3.63)	483 (92.18) 20 (3.82) 21 (4.01)	471 (89.89) 36 (6.87) 17 (3.24)
**Tachypnoea** No Yes Unknown	986 (94.08) 24 (2.29) 38 (3.63)	497 (94.85) 2 (0.38) 25 (4.77)	489 (93.32) 22 (4.20) 13 (2.48)
**Odynophagia** No Yes Unknown	690 (65.84) 242 (23.09) 116 (11.07)	359 (68.51) 114 (21.76) 51 (9.73)	331 (63.17) 128 (24.43) 65 (12.40)
**Congestion** No Yes Unknown	535 (51.05) 479 (45.71) 34 (3.24)	285 (54.39) 216 (41.22) 23 (4.39)	220 (44.53) 263 (53.24) 11 (2.23)
**Fatigue** No Yes Unknown	692 (66.03) 277 (26.43) 79 (7.54)	315 (60.11) 171 (32.63) 38 (7.25)	377 (71.95) 106 (20.23) 41 (7.82)
**Headache** No Yes Unknown	544 (51.91) 343 (32.73) 161 (15.36)	225 (42.94) 232 (44.27) 67 (12.79)	319 (60.88) 111 (21.18) 94 (17.94)
**Conjunctivitis** No Yes Unknown	994 (94.85) 14 (1.34) 40 (3.82)	488 (93.13) 9 (1.72) 27 (5.15)	506 (96.56) 5 (0.95) 13 (2.48)
**Gastro sympt.** No Yes Unknown	690 (65.84) 328 (31.3) 30 (2.86)	350 (66.79) 154 (29.39) 20 (3.82)	340 (64.89) 174 (33.21) 10 (1.91)
**Abdominal sympt.** No Yes Unknown	833 (79.49) 200 (19.08) 15 (1.43)	421 (80.34) 96 (18.32) 7 (1.34)	412 (78.63) 104 (19.85) 8 (1.53)
**Vomiting** No Yes Unknown	920 (87.79) 128 (12.21) - (-)	479 (91.41) 45 (8.59) - (-)	441 (84.16) 83 (15.84) - (-)
**Diarrhoea** No Yes Unknown	879 (83.87) 168 (16.03) 1 (0.10)	452 (86.26) 72 (13.74) 0 (0)	427 (81.49) 96 (18.32) 1 (0.19)
**Dermatologic** No Yes Unknown	980 (93.51) 28 (2.67) 40 (3.82)	483 (92.18) 16 (3.05) 25 (4.77)	497 (94.85) 12 (2.29) 15 (2.86)
**Rash** No Yes Unknown	1032 (98.47) 16 (1.53) - (-)	517 (98.66) 7 (1.34) - (-)	515 (98.28) 9 (1.72) - (-)
**Adenopathies** No Yes Unknown	726 (69.27) 10 (0.95) 312 (29.77)	345 (65.84) 4 (0.76) 175 (33.40)	381 (72.71) 6 (1.15) 137 (26.15)
**Haemorrhages** No Yes Unknown	930 (88.74) 2 (0.19) 116 (11.07)	456 (87.02) 1 (0.19) 67 (12.79)	474 (90.46) 1 (0.19) 49 (9.35)
**Irritability** No Yes Unknown	569 (54.29) 42 (4.01) 437 (41.70)	292 (55.73) 22 (4.20) 210 (40.08)	277 (52.86) 20 (3.82) 227 (43.32)
**Neurological** No Yes Unknown	1010 (96.37) 6 (0.57) 32 (3.05)	499 (95.23) 4 (0.76) 21 (4.01)	511 (97.52) 2 (0.38) 11 (2.10)
**Shock** No Yes Unknown	974 (92.94) 2 (0.29) 71 (6.78)	489 (93.32) 0 (0) 35 (6.68)	485 (92.56) 3 (0.57) 36 (6.87)
**Absence of taste** No Yes Unknown	754 (71.95) 55 (5.25) 239 (22.81)	386 (73.66) 49 (9.35) 89 (16.98)	368 (70.23) 6 (1.15) 150 (28.63)
**Absence of smell** No Yes Unknown	746 (71.18) 60 (5.73) 242 (23.09)	380 (72.52) 55 (10.50) 89 (16.98)	366 (69.85) 5 (0.95) 153 (29.20)

**Table 3 viruses-14-00063-t003:** Average CV scores for the models trained with the data subset including all ages (architecture 1), with the data subset for ages 0 to 5 years (architecture 2) and with the data subset for ages 6 to 14 years (architecture 3).

**Architecture 1**	**AUROC**	**Precision**	**Sensitivity**	**Specificity**	**F1**
XGB	0.645	0.273	0.631	0.66	0.38
RF	0.644	0.278	0.607	0.680	0.381
SVM	0.627	0.289	0.507	0.747	0.367
MLP	0.567	0.253	0.496	0.637	0.264
LR	0.633	0.267	0.597	0.669	0.369
**Architecture 2**	**AUROC**	**Precision**	**Sensitivity**	**Specificity**	**F1**
XGB	0.577	0.141	0.558	0.596	0.225
RF	0.58	0.14	0.603	0.556	0.226
SVM	0.58	0.145	0.576	0.593	0.231
MLP	0.542	0.128	0.465	0.619	0.198
LR	0.564	0.134	0.562	0.567	0.216
**Architecture 3**	**AUROC**	**Precision**	**Sensitivity**	**Specificity**	**F1**
XGB	0.649	0.38	0.632	0.667	0.474
RF	0.652	0.377	0.663	0.641	0.479
SVM	0.651	0.378	0.653	0.65	0.477
MLP	0.596	0.33	0.588	0.605	0.419
LR	0.635	0.365	0.626	0.645	0.46

**Table 4 viruses-14-00063-t004:** Test scores for each of the fine-tuned best performing models with 95% CI.

Subset	Architecture	AUROC (95% CI)	Precision (95% CI)	Sensitivity (95% CI)	Specificity (95% CI)	F1 (95% CI)
**All ages**	XGB	0.65 (0.62–0.67)	0.66 (0.56–0.73)	0.3 (0.27–0.33)	0.65 (0.55–0.73)	0.41 (0.38–0.43)
**0–5 year**	RF	0.63 (0.55–0.67)	0.15 (0.13–0.19)	0.65 (0.43–0.9)	0.59 (0.38–0.74)	0.25 (0.2–0.29)
**6–14 year**	RF	0.67 (0.64–0.69)	0.36 (0.31–0.41)	0.66 (0.56–0.75)	0.68 (0.58–0.79)	0.46 (0.43–0.49)

## Data Availability

Data will be freely available upon request and the link to the replication codes and archived datasets can be found in https://github.com/chus-chus/cov19-modeling (accessed on 16 November 2021).

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
