# Peer review of "Symptom-Based Predictive Model of COVID-19 Disease in Children"

_viruses, 2021, doi:10.3390/v14010063_

Round 1

Reviewer 1 Report

I read with interest the paper entitled "Symptom-based predictive model of COVID-19 disease in children" presented for Review. The paper is innovative and interesting to read; I have however a major concern, and precisely about the use of Rapid Antigenic Tests in Covid diagnosis. The majority of tests administered in the study are rapid and not PCR tests; I suggest a clear explanation of the sensitivity of these tests is advisable, also in the light of the results, where you state that patients who underwent a double test (rapid antigenic and PCR) have a discordance rate of around 30%. 

Beside that, I would suggest a shortening of the tables which occupy a good amount of space and are in this way not well-readable; maybe they could be transposed into a graphic, at least partially?

Author Response

Dear reviewer,

Thank you very much for your comments and suggestions.

Below, you can find in red letter our answers to your questions:

Point 1: I read with interest the paper entitled "Symptom-based predictive model of COVID-19 disease in children" presented for Review. The paper is innovative and interesting to read; I have however a major concern, and precisely about the use of Rapid Antigenic Tests in Covid diagnosis. The majority of tests administered in the study are rapid and not PCR tests; I suggest a clear explanation of the sensitivity of these tests is advisable, also in the light of the results, where you state that patients who underwent a double test (rapid antigenic and PCR) have a discordance rate of around 30%.

Response 1: First of all, thank you very much for your comment. It is in fact a limitation of our study and we hope to have addressed it properly in the Limitations section, where it has been added. The Rapid Antigenic Tests we use have been, in fact, found to have a ≥80% sensitivity and ≥97% specificity when performed to symptomatic patients, especially in the early stages of the symptomatology. This is a requirement that our cohort satisfies, since tests were only performed to symptomatic patients and they were taken at the time of diagnosis. Moreover, the rapid antigenic tests are the first tests used for SARS-CoV-2 diagnosis in all the Primary health-care centers, therefore the most appropriate tests for evaluating the model in a standard-of-care approach.   

Point 2: I would suggest a shortening of the tables which occupy a good amount of space and are in this way not well-readable; maybe they could be transposed into a graphic, at least partially?

Response 2: Thank you again for your advice. It has been of great help. We have been thinking about how to join together some of the tables or graphic the information. Regarding the first option, we think that presenting as much information in one table (such as joining together tables 2, 3 and 4) would be confusing. Besides, since the data gathered is so diverse and complete we reckon that doing bar graphs or query charts would be tedious since a lot of figures would come out from that and it would not help the comprehension of the information. For those reasons, we decided to remove tables 3 and 4 from the main text and move them to the Supplementary material, significantly reducing the “noise” in the main text.

Reviewer 2 Report

Thank you for the opportunity to review this submission.
Changes are needed both in the way the data is presented and in the organization of the article (in its current form it is difficult to follow and understand).

Major changes
1. The way the data set is presented is contradictory/conflicting:

  • at 2.3. Data sources and setting it is mention that ʺwe collected demographic, epidemiological, clinical, and microbiological data on 4,456 symptomatic children with suspected SARS-CoV-2 infection (see Supplementary Information) ʺ.
  • at 3.1. Data description it is mention that ʺA total of 4,456 children were recruited in our study, 44.5% (1984/4456) were female and 42% (1872/4456) younger than 6 years of age, with no significant differences between COVID-19 and non-COVID-19 cases (Table 1)ʺ.

But in Table S1 the number of cases/children is not 4456 (it is 3117, including unknown cases).

  1. In Table 1 in case of X-Ray characteristic the are 77 cases with cu X-Ray performed, 4251 with X-Ray not performed. What is the situation with the difference of 128 cases (up to 4456)?

The same situation is encountered in case of others characteristics from Table 1.

  1. In Table 1 in case of co-viral infection characteristic it is mention a number of 14 cases while in section 3.1. Data description, when these co-infections are described, the number of cases is higher.

  1. Please define what has been included in bacterial infections (Table 1).

  1. Please define what has been included in comorbidities (Table 1).

  1. The characteristics: use school bus, play sports, smokers at home are usually/common features mentioned/specified in the patient's file?

  1. Also, there is contradictory data presentation in Figure 2. For example:
  • the fever symptom is described for 4410 cases (not 4456).
  • the symptom highest fever in missing for 1926 cases while the symptom fever is missing only for 96 cases.

  1. The presented clinical information/symptoms are recorded at the time of hospitalization/testing or it includes also symptoms that developed during the whole course of the infection/hospital stay?

  1. Models related questions:
  • in section 8. Pre-processing it is mention that ʺSymptoms with a percentage of missing data in excess of 25% were eliminated from the models, as explained in the Supplementary Informationʺ, while in supplementary information file at section dealing with missings it is mention that ʺ We have used the IterativeImputer class from scikit-learn [39], which iteratively imputes each feature as a function of the othersʺ;… ʺIn each CV fold the missing values in training and validation are imputed in a round-robin fashion for 5 rounds with a k-nearest-neighbour classifier (k=5)ʺ respectively (in section model development).
  • for a better understanding the authors should mention which parameters/characteristics were selected to be included in each of the developed models (which were the uninformative variables).

  1. The authors should include a data flow diagram in which to highlight the data dynamics for the various stages (pre-processing, modeling).

  1. This is a prospective study?

Minor changes:

  1. Please define the abbreviations RVI (from section 2.3.)
  2. Tables 2, 3 and 4 could be merged, to make it easier to follow/understand.
  3. Table 1 and figure 2 should rather be included in the supplementary material, so as not to make the article difficult to follow (specially for medical practitioners not very familiar with needs/requirements and methods of data pre-processing and analysis).
  4. Please follow the guideline for references (nr. 31-34, 36-38, 43-45).

Author Response

Dear reviewer,

Thank you very mcuh for your comments and suggestions, they have been very helpful to improve the manuscript.

Below, you can find in red letter our responses to your questions.

Major changes

Point 1. The way the data set is presented is contradictory/conflicting: at 2.3. Data sources and setting it is mention that ʺwe collected demographic, epidemiological, clinical, and microbiological data on 4,456 symptomatic children with suspected SARS-CoV-2 infection (see Supplementary Information) ʺ. at 3.1. Data description it is mention that ʺA total of 4,456 children were recruited in our study, 44.5% (1984/4456) were female and 42% (1872/4456) younger than 6 years of age, with no significant differences between COVID-19 and non-COVID-19 cases (Table 1)ʺ. But in Table S1 the number of cases/children is not 4456 (it is 3117, including unknown cases).

Response 1: Firstly, thank you very much for your comments. In line with your (and the other reviewer's) general sentiments over the lengthy tables and how they clutter the document, we have decided to remove Table S1 and move Tables 3 and 4 to Supplementary Information. We firmly believe that Tables 1 and 2 present necessary information that enables the reader to properly follow the article. Tables 3 (now S1) and 4 (now S2) are also of great importance for the more technical part, but they do not have as much added value in the main text and we think that moving them to the Supplementary material is less distracting and enables the reader to follow along more fluidly. On another note, Table S1 is not essential for the description and understanding of the study and we have seen that it can be misleading, for that reason it has been removed. With these changes, we hope that the manuscript is now easier to follow and that the supplementary files include all but not more of the necessary information required for us to properly communicate to the reader the objectives of the present study.

Point 2: In Table 1 in case of X-Ray characteristic there are 77 cases with X-Ray performed, 4251 with X-Ray not performed. What is the situation with the difference of 128 cases (up to 4456)? The same situation is encountered in case of other characteristics from Table 1.

Response 2: The difference in these 128 cases are due to missing values, patients for which this variable was unfulfilled. With the aim of doing a shorter table, we decided to put the number of patients reporting the variable next to its name. Hence, there are in fact 4328 patients that had the X-Ray feature specified (4251 + 77). The same is happening for other variables in this table.

Point 3: In Table 1 in case of co-viral infection characteristic it is mentioned a number of 14 cases while in section 3.1. Data description, when these co-infections are described, the number of cases is higher.

Response 3: Thank you very much for the comment. In Table 1 we comment about 14 co-infections, meaning that 14 individuals presented two or more viruses at the same time, but not necessarily one of them was SARS-CoV-2, since only 4 of these patients tested positive for SARS-CoV-2 as specified in the table. 3 of those 4 patients are described in section 3.1., in which it is stated that “co-viral infections with SARS-CoV-2 were two rhinovirus and one case of enterovirus.” Studying co-infections was not the aim of this study, however we wanted to add this data to inform the readers about this issue.

Point 4: Please define what has been included in bacterial infections (Table 1).

Response 4: A bacterial infection was defined as any case testing positive for specific bacterial PCR or growing in a culture (hemoculture or other specimen cultures), not cases with clinical suspected bacterial infection. We have not been more specific in this field because this is not one of the objectives of the present study.

Point 5: Please define what has been included in comorbidities (Table 1).

Response 5: We were missing some important information in Section 2.7., definitely, and we hope we have addressed that properly at the end of the section. The comorbidities considered have been congenital cardiopathies, hypertension, asthma, chronic pulmonary diseases, renal, liver or neurologic diseases; diabetes mellitus, tuberculosis, primary and secondary immunodeficiencies (including human immunodeficiency virus (HIV) infections), onco-hematological diseases, kawasaki syndrome, auto-inflammatory diseases, obesity, prematurity and palivizumab administration (due to the prematurity). We added an “other comorbidities” check in the questionnaire for very specific and rare conditions.

Point 6: The characteristics: use school bus, play sports, smokers at home are usually/common features mentioned/specified in the patient's file?

Response 6: Thank you for your comment. These characteristics are not something that is usually reported in a patient’s file, but for this study all the researchers (pediatricians) filled a digital questionnaire in the REDCap database. We hope to have clarified it properly at the end of Section 2.7.

Also, there is contradictory data presentation in Figure 2. For example:

Point 7: The fever symptom is described for 4410 cases (not 4456).

Response 7: We hope to have addressed it properly in the main text, in the beginning of Section 2.7., as it required a better explanation. The number of patients used in the chi-squared test is 4419, because we take the subset that undertook a COVID-19 test and had a reported result. Thus, the fever and other symptoms are, indeed, only described for 4419 patients.

Point 8: The symptom highest fever is missing for 1926 cases while the symptom fever is missing only for 96 cases.

Response 8: Each symptom is reported independently, meaning that the fever symptom question was fulfilled for all but 96 patients, but the highest fever symptom was fulfilled for all but 1931 cases. This means that, for some patients, the fever question was answered but not the highest fever question, probably because it was unknown or not registered. The same happens with other “related” symptoms.

Point 9: The presented clinical information/symptoms are recorded at the time of hospitalization/testing or it also includes symptoms that developed during the whole course of the infection/hospital stay?

Response 9: Thank you for your comment. We hope to have clarified the answer in the main text, when we present the data collected at the end of Section 2.3. The clinical information was collected at the time of testing or when the child was diagnosed. With this study, we intend to provide a machine learning technique to be useful at the time of diagnosis.

Models related questions:

Point 10: in section 8. Pre-processing it is mention that ʺSymptoms with a percentage of missing data in excess of 25% were eliminated from the models, as explained in the Supplementary Informationʺ, while in supplementary information file at section dealing with missings it is mention that ʺ We have used the IterativeImputer class from scikit-learn [39], which iteratively imputes each feature as a function of the othersʺ;… ʺIn each CV fold the missing values in training and validation are imputed in a round-robin fashion for 5 rounds with a k-nearest-neighbour classifier (k=5)ʺ respectively (in section model development).

Response 10: As you pointed out, we believe it is necessary to make this more clear to the reader: only the features / symptoms with less than 25% of missing values go through the imputation step, as the ones with more than 25% of missing values were previously discarded. We have made the appropriate changes in the “Dealing with missings” section of Supplementary Information to reflect this issue.

Point 11: for a better understanding the authors should mention which parameters/characteristics were selected to be included in each of the developed models (which were the uninformative variables).

Response 11: Thank you for the suggestion. We would like to point out that tables 2, 3 and 4 contain all of the symptoms that have been included in each model (respectively: the model for pediatric aged patients, for patients aged from 0 to 5 and for patients from 6 to 15). In section 2.8, at the end of the 4th paragraph, we stated the following:“Tables 2, 3 (now table S1) and 4 (now table S2) show the final variables that were chosen to train the three final models.”. To make the point clearer, we have replaced “variables” with “symptoms”.

Point 12: The authors should include a data flow diagram in which to highlight the data dynamics for the various stages (pre-processing, modeling).

Response 12: We hope to have addressed it properly in the main text of the article. We have changed Figure 1 to be a more complete and comprehensive flowchart of our pipeline. It now shows how the data has been used / manipulated in each of the modeling steps, as is thoroughly described in Supplementary Information.

Point 13: This is a prospective study?

Response 13: Thank you for the question. As stated in section 2.1. Study design, “this is a prospective, observational, cross-sectional study of symptomatic children <16 years-old with a suspected COVID-19 disease, who were tested with rapid antigenic diagnostic tests,  or reverse transcription polymerase chain reaction (RT-PCR) or both for confirmatory diagnosis in any of the participating Primary or Hospital health-care centers between 1 November 2020 and 31 March 2021, following the standard of care for diagnosing these cases.” We follow the patients’ condition from the moment they go to see their pediatrician to when they have a diagnosis after the results of the tests performed to the patients come out. We would like to express again our gratitude for your extensive review, it has been very helpful.

Minor changes:

Point 14: Please define the abbreviations RVI (from section 2.3.)

Response 14: Thank you for your advice, we have addressed it.

Point 15: Tables 2, 3 and 4 could be merged, to make it easier to follow/understand.

Response 15: Thank you for the suggestion. It has been of great help. We have been thinking about how to join together some of the tables or graphicalize the information. Regarding the first option, we think that presenting as much information in one table (such as joining together tables 2, 3 and 4) would be confusing, since each table represents a different subset of patients. Besides, since the data gathered is so diverse and complete we reckon that doing bar graphs or query charts would be tedious since a lot of figures would come out from that and it would not help the comprehension and compression of the information. For those reasons, we decided to remove tables 3 and 4 from the main text and move them to the Supplementary material, significantly reducing the “noise” in the main text.

Point 16: Table 1 and figure 2 should rather be included in the supplementary material, so as not to make the article difficult to follow (specially for medical practitioners not very familiar with needs/requirements and methods of data pre-processing and analysis).

Response 16: Thank you for the question. We have changed Figure 2 location and it is now in the Supplementary Materials section. We agree that it makes the article easier to follow. However, we have misgivings about moving Table 1 to the Supplementary information, since we think that it is important for it to be in the main text, for the readers to know the diversity of the data gathered.

Point 17: Please follow the guideline for references (nr. 31-34, 36-38, 43-45).

Response 17: Thank you for your advice, we apologize for the inconvenience. We hope we have addressed it properly.

Round 2

Reviewer 1 Report

Dear Authors,

I have read your reply and your corrections to the paper and the results satisfy the objections I posed, so that no more modification is required in my opinion and the paper can be published.

Good luck and cogratulations!

Reviewer 2 Report

Yes, the manuscript has been sufficiently improved to were publication in Viruses.